# Inhibition of Citric Acid-Induced Dentin Erosion by an Acidulated Phosphate Sodium Monofluorophosphate Solution

**DOI:** 10.3390/ma16155230

**Published:** 2023-07-25

**Authors:** Ryouichi Satou, Susumu Ueno, Hideyuki Kamijo, Naoki Sugihara

**Affiliations:** 1Department of Epidemiology and Public Health, Tokyo Dental College, Tokyo 101-0061, Japan; sugihara@tdc.ac.jp; 2Department of Occupational Toxicology, Institute of Industrial Ecological Sciences, University of Occupational and Environmental Health, Fukuoka 807-8555, Japan; susumu@med.uoeh-u.ac.jp; 3Department of Social Security for Dentistry, Tokyo Dental College, Tokyo 101-0061, Japan; kamijohideyuki@tdc.ac.jp

**Keywords:** dentin, erosion, fluoride, demineralization, preventive dentistry

## Abstract

Sodium monofluorophosphate (Na_2_FPO_3_, MFP) is mainly used as an ingredient in fluoride-based dentifrices as it has a high safety profile, with one-third of the toxicity of sodium fluoride (NaF), as well as the ability to reach deep into the dentin. The purpose of this study was to assess the prevention of dentin erosion by MFP upon exposure to citric acid, which has a chelating effect, and to compare the effects to those of the conventional acidulated phosphate fluoride (APF) application method. Bovine dentin was used, and four groups were created: (i) APF (9000 ppmF, pH 3.6) 4 min group; (ii) acidulated phosphate MFP (AP-MFP, 9000 ppmF, pH 3.6) 4 min group; (iii) AP-MFP 2 min + APF 2 min (dual) group; and (iv) no fluoride application (control) group. Compared with the conventional APF application method, the application of AP-MFP was shown to significantly reduce substantial defects, mineral loss, and lesion depth; better maintain Vickers hardness; and promote the homogenous aggregation of fine CaF_2_ particles to seal the dentin tubules, enhancing acid resistance in their vicinity. The ΔZ value of the AP-MFP group was 2679 ± 290.2 vol% μm, significantly smaller than the APF group’s 3806 ± 257.5 vol% μm (*p* < 0.01). Thus, AP-MFP-based fluoride application could effectively suppress citric acid-induced demineralization and could become a new, more powerful, and biologically safer professional-care method for preventing acid-induced dentin erosion than the conventional method.

## 1. Introduction

The recent worldwide surge in health consciousness has led to many alterations in dietary habits, and one of the effects of this phenomenon is a prominent escalation of dental erosion across a broad spectrum of age groups, from young to older individuals. This can be attributed to various factors, such as a shift towards vegetarianism, the increase in consumption of citrus fruits, and the habitual intake of beverages containing citric acid and phosphoric acid [1,2]. Dental erosion is defined as the demineralization of teeth through chemical processes unrelated to microbial activity [1]. Although several acids, including lactic, phosphoric, acetic, and citric acids, have been implicated in dental erosion, it is well established that the demineralization potential of citric acid, which possesses chelating properties, is significantly higher than that of other acids [1,2,3]. Citric acid is predominantly found in citrus fruits and is consumed frequently nowadays, and prolonged and habitual intake of citrus fruits or citrus-based beverages is known to have the potential to lead to severe dental erosion [3]. 

Furthermore, most edentulous older individuals over 65 years of age are reported to have physiological or pathological gingival recession and therefore exposed root dentin in the oral cavity [4]. Dentin has a higher collagen content than enamel, as well as a higher critical pH range (6.0–6.2), making it more susceptible than enamel to the detrimental effects of acids [4]. Consequently, dentin erosion has a faster progression and wider extent than that of enamel, emphasizing the need for its early and effective management in clinical dentistry. 

Although the primary preventive measure for dental erosion is the enhancement of the acid resistance of the tooth structure through the application of fluoride, it is worth noting that most previous studies evaluating the effectiveness of this approach have focused on the enamel. As a result, there is limited evidence regarding the prevention of dentin erosion [5], as well as valid concerns that these preventive measures may not provide sufficient protection against it, particularly in cases of exposure to substantial acid levels beyond the buffering capacity of saliva within a short period of time [5,6,7]. Toothpastes containing fluoride have not proven effective in preventing erosion, and no significant correlations were found between the type of toothbrush used and its frequency of use in 231 children [6]. Therefore, the development of preventive methods that specifically target dentin erosion is urgently needed.

Monofluorophosphate sodium (MFP) is a fluoride compound primarily used as a therapeutic ingredient in toothpastes formulated for self-care purposes [8]. No significant differences in caries prevention efficacy have been observed between toothpastes containing MFP and those containing sodium fluoride (NaF) [9]. Nevertheless, it has been established that toothpastes containing MFP promote demineralization inhibition and remineralization of both enamel and dentin [10]. One more advantage of MFP from the perspective of clinical applications is its significantly low systemic toxicity, which can be attributed to its containing fluoride ions in a complex state. According to a report published by White et al. in 1983, the toxicity of MFP was approximately one-third that of NaF [11,12]. MFP is also quite soluble and is reported to result in a three-fold higher fluorine ion concentration than NaF when considering the molar mass of fluorine atoms in solution [12]. In the presence of a substantial calcium ion concentration in solution, the fluoride ions of NaF readily react to form calcium fluoride (CaF_2_), leading to a significant reduction in the fluoride ion concentration in the tooth-surface microenvironment [13]. However, MFP is capable of preserving the structure of the fluoride ion complex even in solutions containing calcium, which can result in fluoride ion concentrations 70 times higher than those in solutions with NaF [13]. Furthermore, a significant advantage of utilizing MFP in oral care products and within the oral cavity is its property of not reacting with saliva or even with solutions supersaturated with calcium ions [14]. Notably, when MFP reacts with tooth enamel, it exhibits lower immediate effectiveness and inferior near-surface-layer (0–50 μm) acid resistance compared to NaF [13]; however, due to differences in the mechanisms of action of MFP and NaF, MFP has the potential to penetrate not only the surface layer but also the deeper regions (50–300 μm) of the tooth enamel to form a thick and uniform acid-resistant layer [12,15,16]. 

Nevertheless, despite having several advantages over NaF, the application of MFP remains limited to self-care toothpastes with a maximum concentration of 1500 ppm [8]. However, by using MFP at higher concentrations, it would be possible to develop highly biocompatible, professional-care tooth-surface application methods for preventing dental erosion that can also effectively target the deeper dentin layer. The objective of this study was to leverage the superior biocompatibility of MFP and its ability to impact the deeper layers of the dentin in order to develop a novel strategy to prevent dental erosion. Specifically, we also aimed to evaluate its effectiveness in enhancing the resistance of the dentin to citric acid by comparing it with conventional preventive treatment methods. 

## 2. Materials and Methods

### 2.1. Preparation of Dentin Samples

A set of 36 bovine mandibular incisors was used in this study. Subsequently, dentin samples (dimensions: 1 cm × 1 cm × 1 cm) were meticulously fashioned and subjected to a thorough polishing process using water-resistant abrasive paper (grit sizes: #1000, #2000, and #4000), resulting in a mirror-like surface finish.

### 2.2. Fluoride Application and pH-Cycling Acid Challenge

The samples were divided into four groups: (1) control group (no fluoride application); (2) acidulated phosphate fluoride (APF) (9000 ppmF, pH 3.6) 4 min group; (3) AP-MFP solution (9000 ppmF, pH 3.6) 4 min group; and (4) AP-MFP 2 min + APF 2 min dual group (*n* = 9 samples in each group). In order to establish both an experimental and control window on the same surface, dental wax (Inlay Wax Soft, 27B2X00008000028; GC Co. Ltd., Tokyo, Japan) was applied to half of each mirror-polished dentin surface. 

A pH-cycling test was subsequently performed using a Stefan curve-based protocol as described by Matsuda et al. [17,18,19]. Each cycle consisted of exposure to a pH of 5.5 or less (continuous pH change with a minimum pH of 4.0 due to citric acid demineralization solution, average duration: 37 ± 5 min) and recovery to pH 7.3 (continuous pH change from pH 5.5 to pH 7.3 with addition of remineralization solution, duration: 23 ± 3 min), with the total average duration from the start of the cycle to the return to the initial pH value of 7.3 being 60 ± 5 min. After fluoride application, all samples were immersed in a remineralization solution (0.02 M HEPES buffer solution, Ca: 1.5 mM, P: 0.9 mM, pH 7.3, degree of saturation: 5.5) for 1 h at 37 °C. After the remineralization treatment, samples were immersed in a demineralization solution (0.1 M citric acid buffer solution, pH 4.5) for 37 ± 5 min of pH cycling at 37 °C. Citrate buffer solution was prepared by mixing 6.72 g of citric acid (7447-40-7, Fujifilm wako, Osaka, Japan) and 4.41 g of sodium citrate (68-04-2, Fujifilm wako, Osaka, Japan) in 1000 mL. HEPES buffer was composed of 0.1M CaCl_2_ (10043-52-4, Fujifilm wako, Osaka, Japan) 15 mL, 0.1M KH_2_PO_4_ (7778-77-0, Fujifilm wako, Osaka, Japan) 9 mL, KCl (7447-40-7, Fujifilm wako, Osaka, Japan) 9.69 g, and HEPES (7365-45-9, Fujifilm wako, Osaka, Japan) 4.77 g, mixed up to 1000 mL and adjusted to pH 7.3. Each cycle (fluoride application, remineralization, and demineralization treatment) was repeated ten times.

### 2.3. Three-Dimensional Laser Microscopy

After wax removal, the samples were dehydrated using an ascending ethanol series. For evaluating the differences in step height profile between the experimental surface (ES) and reference surface (RS) following the pH-cycling procedure, a three-dimensional (3D) measurement laser microscope (LEXT OLS4000; Olympus Corp., Tokyo, Japan) was employed. Tooth defects resulting from the acid challenge within a 645 µm × 645 µm area were assessed, with photographs taken at the boundary between the acid-demineralized ES and wax-protected RS. The samples were also analyzed to determine the average roughness (Sa) of the ES in the 645 µm × 645 µm area. Wavelengths >80 µm were excluded from the cross-sectional curve to obtain the roughness curve. The number of substantial defects and Sa were measured at five distinct points per sample at the boundary between the ES and RS, and the mean and standard deviation (SD) values were calculated.

### 2.4. Vickers Hardness Measurement 

After dehydration, the Vickers hardness of the samples was measured using a hardness tester (HMV-1; Shimadzu Corp., Tokyo, Japan) and an indentation load and time of 0.49 N and 20 s, respectively. To account for individual sample differences, the change in hardness before and after the experiment (∆HV = RS hardness − ES hardness) was calculated. Vickers hardness and ∆H values were recorded at five locations per sample, and the mean and SD values were calculated.

### 2.5. Cross-Sectional Morphology Assessment Using Scanning Electron Microscopy (SEM)

Following pH cycling, the surface of each sample was rinsed with xylene and then subjected to carbon vapor deposition. Thereafter, the tooth surfaces were examined using a scanning electron microscope (SU6600; HITACHI Ltd., Tokyo, Japan) at an accelerating voltage of 15 kV. 

### 2.6. Contact Microradiography (CMR)

The imaging conditions and analysis methodology employed in this study were based on those used in a previous study by Sato et al., with reference to Angmar’s formula [20,21]. To prepare polished, 100 µm thick sections, the samples were embedded in a polyester resin (Rigolac; Nisshin EM, Tokyo, Japan). Soft X-ray imaging was conducted using a 20 µm Ni filter, while light microscopy at 200× magnification was performed using a glass plate (High Precision Photo Plate, HRP-SN-2; Konica Minolta, Tokyo, Japan). The following parameters were used for imaging using the CMR-3 system (Softex, Tokyo, Japan): tube voltage, 15 kV; tube current, 3 mA; radiation time, 8 min. Subsequently, the acquired images were analyzed using the Image Pro Plus software (version 6.2; Media Cybernetics Inc., Silver Spring, MD, USA) and an image analysis system (HC-2500/OL; OLYMPUS Corp., Tokyo, Japan) to obtain the concentration profiles. Mineral loss (ΔZ) and lesion depth (Ld) were determined to compare the extent of demineralization. The values were converted to a histogram with a mineral value of 0% and a healthy enamel section of 100%. Ld was defined as the distance from the enamel surface to the location of a lesion where the mineral content was greater than 95% of that in the sound enamel. To calculate the ΔZ value for each specimen from the profiles, the area under the curve was subtracted from the assumed area of the sound enamel before demineralization. 

### 2.7. Statistical Analysis 

The statistical significance of the results was determined using Kruskal–Wallis one-way analysis of variance (threshold value, *p* < 0.01). The Bonferroni test was used for subsequent post hoc comparisons. The Origin 2023 software (Lightstone Corp., Tokyo, Japan) was used for data analysis and graph generation.

## 3. Results

### 3.1. Step Height Profiles after pH Cycling

Figure 1 depicts the surface profile measurements obtained using 3D laser microscopy after pH cycling. In Figure 1a–d, the RS, which was not demineralized and was protected using wax, can be seen on the left, whereas the demineralized ES can be seen on the right. In the control group, the ES was significantly demineralized (mean defect size on the dentin surface: 23.157 ± 2.290 μm; Figure 1a,e). In the APF group, the difference in height between the RS and ES was much lower (7.531 ± 1.885 μm), and there was a significant inhibition of demineralization compared with that in the control group (*p* < 0.001) (Figure 1b,e). The AP-MFP group had an even smaller height difference (7.482 ± 0.941 μm) compared with that in the APF group. Among the four groups, the extent of dentin demineralization was the lowest in the AP-MFP group (Figure 1c,e). The difference in the dual group was 8.356 ± 1.065 μm, which was significantly more than those in the APF and AP-MFP groups (Figure 1d,e). There were no significant differences among the APF, AP-MFP, and dual groups (*p* > 0.05, Figure 1e).

### 3.2. Calculated Average Roughness after pH Cycling 

The calculated average roughness results are presented in Figure 2, with the mean values indicated using gray squares, the medians as horizontal lines, the lower quartiles as the lower limits, and the upper quartiles as the upper limits. Significant irregularities on the dentin surface were observed in the control group (mean Sa: 0.295 ± 0.021 μm, median: 0.295 μm [0.283–0.312]), and there were significant differences between the control group and other groups (*p* < 0.001). The Sa value was highest in the AP-MFP group (mean: 0.381 ± 0.027 μm, median: 0.383 μm [0.351–0.409]); however, no significant difference was observed compared with that in the APF group (mean: 0.339 ± 0.038 μm, median: 0.334 μm [0.310–0.377]; *p* > 0.05). The Sa value was lowest in the dual group (mean: 0.250 ± 0.016 μm, median: 0.247 μm [0.236–0.267]) and was significantly different from those in all the other groups (*p* < 0.001).

### 3.3. Vickers Hardness and Changes after pH Cycling 

Figure 3a shows the results of the Vickers hardness analysis for the demineralized surfaces in each experimental group. The mean Vickers hardness value in the control group was 19.889 ± 2.004 (median: 20.713 [18.654–21.936]), which was the lowest among all the groups. The Vickers hardness was higher in the APF group compared with that in the control group (mean: 28.397 ± 1.707, median: 28.342 [26.801–29.333]), and significant differences were observed compared with those in the other groups as well (*p* < 0.01). The Vickers hardness was highest in the AP-MFP group (mean: 33.041 ± 2.123, median: 33.224 [30.811–35.124]), and there was a significant difference between the APF and AP-MFP groups (*p* < 0.01). The hardness in the dual group was similar to that in the AP-MFP group (mean: 32.736 ± 1.617, median: 32.729 [31.476–33.989]), with no significant difference between the two (*p* > 0.05).

Figure 3b shows the delta HV data, which indicate the changes in Vickers hardness before and after pH cycling. The mean change in Vickers hardness was highest in the control group (25.538 ± 5.771, median: 23.299 [21.786–28.328]), and it was significantly different compared with the changes in the other groups (*p* < 0.01). The mean value in the APF group was 15.308 ± 2.315 (median: 14.781 [14.011–16.811]), demonstrating a significant decrease compared with that in the control group. The smallest change in Vickers hardness was observed in the AP-MFP group (mean: 10.050 ± 2.971, median: 9.557 [7.969–12.685]). However, there was no significant difference between the AP-MFP and APF group values. The change in the dual group was similar to that in the AP-MFP group (mean: 11.665 ± 2.949, median: 11.734 [9.194–14.342]), with no significant difference between the two (*p* > 0.05).

### 3.4. Dentin Surface SEM Observations after pH Cycling

Representative SEM images of the demineralized dentin surface after pH cycling are shown in Figure 4. The control group exhibited an expansion of the dentinal tubule orifices due to demineralization by citric acid; moreover, no particle formation was observed in the surrounding dentin tubes and dentin matrix (Figure 4a,e). Partial closure of the dentinal tubules and aggregates of particulate matter within the tubules were observed in the APF group (Figure 4b,f). Additionally, fine spherical particles could be observed adhering to the surface of the dentin surrounding the tubules (Figure 4f). The AP-MFP group exhibited findings similar to the APF group, with closure of the tubular orifices and the presence of fine particles adhering to the surface; however, the particles observed were noticeably distinct, as spherical particles of a larger diameter were identified (Figure 4c,g). When observed at high magnification, these particles appeared as secondary aggregates composed of finer particles and were found not only in the surrounding dentin tubes but also penetrating into the tube interior (Figure 4g). The dual group exhibited a smooth texture with few overall irregularities, and, as in the APF group, fine adherent particles were observed (Figure 4d,h). The closure of tubule orifices was almost complete, and a few large secondary particles, as observed in the AP-MFP group, were found in the dentin tubules (Figure 4h).

### 3.5. Cross-Sectional SEM Imaging after pH Cycling

Representative reflected electron microscopy images of the cross-sectioned demineralized regions after pH cycling are shown in Figure 5. In the control group, a region with decreased signal intensity, indicative of a reduction in dental calcium density, was observed approximately 15–20 μm from the surface due to demineralization by citric acid (Figure 5a,e). Lateral expansion of dentin tubules was also observed in the area affected by surface demineralization (Figure 5e). In the APF group, a gradual decrease in signal intensity was observed approximately 20–30 μm from the surface, creating a gradient-like pattern (Figure 5b,f). Notably, a significant decrease in signal intensity was observed localized around the dentinal tubules, indicating progressive demineralization centered around the tubules (Figure 5f). The overall signal intensity in the APF group was higher compared with that in the control group, although there was a recovery of signal intensity in the superficial 2–5 μm deep layer (Figure 5f). In the AP-MFP group, the decreased signal intensity was localized to the superficial 5–10 μm, in contrast to that in the other groups, and the signal intensity beyond 10 μm remained uniform (Figure 5c,g). Particles were observed within the dentinal tubules, causing their closure, and no demineralization pattern was observed around the tubules (Figure 5g). The dual group exhibited a gradient-like decrease in signal intensity at 20–30 μm from the surface, similar to that in the APF group (Figure 5d,h). Under 5000-fold magnification, a reduction in signal intensity around the dentinal tubules and a lateral expansion of the tubules themselves were observed in the dual group, with the decreased signal intensity extending along the tubules (Figure 5h).

### 3.6. Measurement of Mineral Loss and Lesion Depths Using CMR Analysis

Figure 6 shows CMR images of dentin cross-section after pH cycling and the variations in mineral content (vol% μm) with respect to depth in each group. In the control group, a region with low signal intensity was observed in the surface layer of the dentin at a depth of 25–35 μm, and a rise in the curve was observed at around 30 μm (Figure 6a,e). In the APF group, the rise in the curve was at a shallower depth (20–25 μm from the surface of the tooth) compared with that in the control group (Figure 6b,e). In the AP-MFP group, the mineral content was over 80% in the 20–25 μm depth range, with the content at the shallowest depth being the highest among the four groups (Figure 6c,e). The mineral loss characteristics in the dual group were similar to those in the APF group, with the graph rising shallower than in the APF group but with a lower slope (Figure 6d,e).

Figure 7 shows the mineral loss (ΔZ, vol% μm) and lesion depth (Ld, μm) values in each group as assessed using CMR. The ΔZ value in the control group was 5985 ± 319.9 vol% μm, which was significantly greater than those in all the other groups (*p* < 0.01, Figure 7a). The APF group showed a reduction in the ΔZ value to approximately two-thirds of that in the control group, with a value of 3806 ± 257.5 vol% μm. The AP-MFP group had the lowest ΔZ value (2679 ± 290.2 vol% μm), approximately half of that in the control group. A significant difference was observed between the APF and AP-MFP groups (*p* < 0.01; Figure 7a). The value in the dual group was similar to that in the APF group (3772 ± 376.1 vol% μm). The difference between the AP-MFP and dual groups was significant (*p* < 0.01).

The Ld value was largest in the control group (52.17 ± 9.548 μm), with significant differences observed between the values in the control, APF, AP-MFP, and dual groups (*p* < 0.01, Figure 7b). The Ld value decreased to 36.66 ± 3.572 μm in the APF group and to 24.46 ± 3.562 μm in the AP-MFP group. A significant difference was observed between the APF and AP-MFP groups (*p* < 0.01; Figure 7b). The value in the dual group (36.44 ± 3.713 μm) was similar to that in the APF group and was significantly different compared with that in the AP-MFP group (*p* < 0.01).

## 4. Discussion

### 4.1. Resistance of Dentin to Citric Acid after AP-MFP Treatment

A substantial reduction in dentin loss was observed in the AP-MFP and dual groups, similar to that in the conventional APF group, after citric acid demineralization (Figure 1). These findings align with previous research indicating the capacity of MFP to enhance the acid resistance of hydroxyapatite (HAP) and dentin, as well as with the results of a study that used surface application of AP-MFP to enamel at a concentration of 9000 ppmF [13,22,23]. APF application reduced enamel loss to a more significant extent than AP-MFP application; however, there was no difference in dentin loss between the AP-MFP and APF groups in this study [23]. This suggests that the improvement in acid resistance achieved by AP-MFP may be higher in dentin than in enamel. Compared to enamel, dentin has a porous structure characterized by the presence of tubules and a higher proportion of collagen, and this composition offers several advantages in terms of ion permeability and reaction kinetics [4,24]. These characteristics of dentin structure and composition may explain why AP-MFP showed a higher demineralization inhibitory effect on dentin than on enamel. Previous studies comparing the effects of APF solutions on enamel and dentin also showed that fluoride application on dentin is more effective than on enamel [4]. Taken together, all these results suggest that AP-MFP, similar to APF, effectively inhibits dentin demineralization.

The calculated average roughness increased in both the APF and AP-MFP groups compared with that in the control group, with only the dual group showing a significant decrease (Figure 2). The surface SEM analysis revealed the partial closure of the dentin tubules in the APF group, whereas the closure of the dentin tubule orifices and the presence of secondary particles formed by the aggregation of fine particles were observed in the AP-MFP group (Figure 4c,g). The overall surface texture was smoother and less rough in the dual group (Figure 4h). When the APF solution was applied to dentin, a rapid reaction occurred between fluoride ions and the calcium ions in the tooth structure, forming particles within the surface layer of the tooth and inside the dentin tubules [25,26,27]. Moreover, surface SEM images showed abundant formation of CaF_2_ particles in the APF, AP-MFP, and dual groups (Figure 4). The increased surface roughness in the APF and AP-MFP groups in this study may have been due to both dentin dissolution by citric acid demineralization and the formation of CaF_2_ particles. In particular, the formation of many secondary particles larger than the expected size of CaF_2_ particles was observed in the AP-MFP group, which may have caused the surface roughness in the AP-MFP group to be higher than in the APF group (Figure 2 and Figure 4). It has been shown that APF is superior to AP-MFP in terms of dentine acid resistance in the surface layer of enamel (at 5–10 μm) [23,28]. The surface roughness results in this study may indicate that APF is superior to AP-MFP in terms of acid resistance in the surface layers of dentin and enamel. 

CaF_2_ particles were observed to be smaller in the dual group than in the APF group, whereas the number of secondary particles was lower than that in the AP-MFP group (Figure 4). It has been shown that CaF_2_ particle formation and diameter are influenced by fluoride ion concentration and reaction time [25,26,27]. The dual group had a short APF reaction time of 2 min, which may have resulted in the formation of fewer CaF_2_ particles. Additionally, the AP-MFP group reaction time of 2 min was also insufficient, which is likely to have resulted in the formation of fewer secondary particles. It is speculated that, owing to the deficiency of CaF_2_ particles, most of the particles in the dual group were demineralized after citric acid treatment and did not remain on the surface, thereby reducing surface roughness (Figure 2).

In this study, we conducted a qualitative assessment of acid resistance based on Vickers hardness and cross-sectional SEM observations, as well as a quantitative assessment of mineral loss (ΔZ) and lesion depth (Ld) using CMR (Figure 3, Figure 4, Figure 5, Figure 6 and Figure 7). The Vickers hardness was highest in the AP-MFP group and was significantly higher than that in the APF group (*p* < 0.01, Figure 3a). Vickers hardness is used as a measure of the degree of internal structural damage to dental tissue and is reduced when this internal tooth structure is disrupted, for example, by acids [29]. The extent of the decrease in signal intensity in demineralized areas in the cross-sectional SEM images was smaller in the AP-MFP group than in the APF group, providing supporting evidence for the Vickers hardness results (Figure 3 and Figure 5). In the AP-MFP group, lateral expansion of the dentinal tubules and demineralization in their vicinity were observed, in contrast to the findings in the dual group. However, no demineralization was observed around the tubules in the AP-MFP group, and the extent and depth of demineralization were smaller and shallower, respectively (Figure 5g,h). The enhanced acid resistance in the deeper layers of dental tissue observed in the AP-MFP group is consistent with the findings of MFP permeating the dental tissue and forming an acid-resistant layer at a depth of 100–300 μm [14,28]. The significant inhibitory effect of AP-MFP on demineralization was also quantitatively supported by both ΔZ and Ld values, as these were lowest in the AP-MFP group and showed significant differences compared to those in the APF and dual groups (*p* < 0.01, Figure 7). A study comparing the acid resistance of HAP treated with NaF and MFP at the same concentration demonstrated that the MFP treatment resulted in higher acid resistance than the NaF treatment [15]. Therefore, the results of previous studies and the present study suggest that applying AP-MFP to the tooth surface improves the acid resistance of dentin and inhibits citric acid demineralization better than the conventional APF method.

The main difference between the AP-MFP and dual groups lies in the presence of APF and the reaction time of the solution application. In the AP-MFP group, a single fluoride agent was applied for 4 min, whereas in the dual group, two fluoride agents were applied for 2 min each. Therefore, it was necessary to consider whether the reaction times of APF and AP-MFP were sufficient for each of the dual group results. The recommended reaction time for the APF solution with regard to tooth surface application in dental practice is 4 min [30,31], and fluoride uptake improves with increasing application time, with a 4 min application resulting in approximately 1.2 times the fluoride uptake of a 1 min application [32]. Bruun et al. reported that, based on the amount of CaF_2_ formed, the reaction between NaF and enamel reached a plateau in approximately 1 to 1.5 min [31]. Additionally, Nishida et al. reported that there was approximately 80–90% fluoride ion uptake in dentin (10–200 μm) within 1 min compared to 4 min of application [30]. Therefore, it can be predicted that a 2 min application is sufficient for APF to provide adequate acid resistance. However, there have been no reports on the application time or fluoride ion uptake of MFP at a concentration of 9000 ppmF. APF exhibits immediate effectiveness owing to the formation of CaF_2_ shortly after application [31]. In contrast, the mechanism by which MFP results in acid resistance involves the penetration of the dental tissue by MFP ions, which results in a relatively slower onset of action than that of APF [15,16]. Therefore, it is possible that the 2 min application time of AP-MFP in the dual group was insufficient to provide an adequate effect, as it may not have allowed enough time for an optimal action to occur. Indeed, it can be inferred that the longer MFP application time in the AP-MFP group compared to that in the dual group is the reason why the AP-MFP had improved acid resistance in dentin. The dual group was designed with the expectation of harnessing the effects of APF and AP-MFP through a combined application. However, it is believed that the anticipated effects were not achieved because of insufficient AP-MFP application time on the dental surface in the dual group. This study revealed that the effectiveness of AP-MFP, similar to that of APF, is also dependent on application time. The application time of AP-MFP was set to 4 min to align with the conventional method. However, whether 4 min is the optimal application time for AP-MFP requires further investigation in future studies to establish a clearer understanding.

### 4.2. Dentin Acid Resistance Mechanism of AP-MFP

In the case of tooth-surface coating using a high NaF concentration, a complex dissolution reaction occurs between HAP and fluoride, resulting in the deposition of CaF_2_ and CaF_2_-like deposits (weakly bound fluorides) on the tooth surface [33,34]. When the pH of the oral cavity decreases, CaF_2_ acts as a reservoir, gradually releasing low concentrations of fluoride ions, thereby improving the acid resistance of dental tissue [33,34]. In acidic environments, NaF forms an intermediate product known as dicalcium phosphate dihydrate (CaHPO_4_·2H_2_O), which effectively facilitates the incorporation of fluoride ions into HAP [35]. It has been reported that AP-MFP, similar to NaF, may have mechanisms that promote fluoride ion uptake into enamel in acidic environments [23]. Two hypotheses have been proposed regarding the mechanism by which MFP enhances acid resistance in the dentin when applied to the dental surface. The first hypothesis suggests that MFP is hydrolyzed by phosphatases in the oral cavity, leading to the release of fluoride ions, which then exhibit mechanisms of action similar to those of NaF [36,37]. The second hypothesis proposes that PO_3_F^2−^ ions replace HPO_4_^2−^ in HAP [15,16]. Both hypotheses provide potential explanations for the mechanism of acid resistance improvement in dentin through the dental surface application of AP-MFP. No plaque or saliva samples were used in the present study. However, it is highly probable that phosphatases remain within the dentinal tubules and residual collagen. Therefore, it is likely that the acid resistance enhancements described in both hypotheses were observed in this experimental system. 

There are multiple reports regarding the substances generated on the surface of dentin through the application of MFP. Tanizawa et al. reported the formation of CaF_2_ and CaPO_3_F on HAP surfaces using electron spectroscopy for chemical analysis following the application of a high concentration of MFP (10,000 ppmF) [13]. Yamagishi et al. hypothesized the formation of MFP-Ca salts with lower aqueous solubility and the generation of MFP-modified HAP, suggesting their presence in addition to that of CaF_2_ [14]. In the surface SEM images obtained in this study, large-diameter secondary particles were observed in both the AP-MFP and dual groups (Figure 4). Although no previous studies have reported similar findings regarding these secondary particles, their presence was consistently observed in all samples from both groups. The particle size and amount of CaF_2_ formed after fluoride application increase depending on the fluoride ion concentration and the reaction time [25,26,27]. The presence of MFP enhances the formation reaction of CaF_2_ and promotes the generation of fluorapatite, thereby improving acid resistance and remineralization [28,38,39]. Due to Van der Waals forces, smaller particles tend to aggregate because of their high surface energy and instability [40]. Therefore, it is possible that the fine CaF_2_ particles generated through reaction enhancement by MFP in the AP-MFP and dual groups underwent homocoagulation. This hypothesis was supported by the SEM images from the AP-MFP group, which revealed that the surfaces of the secondary particles consisted of aggregates of smaller particles (Figure 4g). The presence of larger secondary particles, observed exclusively in the AP-MFP-treated group, implies that the CaF_2_ formation reaction was facilitated by MFP, leading to homogenous aggregation and subsequent particle growth.

### 4.3. Limitations of Research Methods and Prospects for Clinical Dentistry

In this study, the efficacy of AP-MFP was experimentally tested using citric acid, which possesses chelating properties and poses a high risk of dental erosion. However, it should be noted that various acids, such as phosphoric acid, acetic acid, hydrochloric acid, and oxalic acid, are known to contribute to dental erosion. Additionally, their dissociation constants, pH levels, and factors such as dietary or occupational sources can vary significantly [1,2]. Therefore, it is necessary to elucidate the ability of AP-MFP to induce dentin resistance to these other acids as well. Furthermore, it should be noted that this study was conducted in an in vitro setting in the absence of saliva and plaque, which may lead to different characteristics in the actual oral environment where saliva proteins and enzymes are abundant. Therefore, further investigations are also warranted to assess the oral stability of AP-MFP in the presence of MFP-degrading enzymes. Future experiments should be conducted with human teeth. Considering the higher biocompatibility of MFP compared to that of NaF in dental clinical practice, AP-MFP is believed to be particularly effective for pediatric patients with a higher fluoride uptake requirement and oral conditions necessitating fluoride ion penetration into the deep layers of dentin, such as white spot lesions. In this study, the application time was set to 4 min based on the conventional method. However, to enhance the effectiveness of the relatively slow-acting MFP, it may be preferable to extend the application time beyond 4 min or apply it in the form of a high-viscosity gel to allow continuous and prolonged contact with the tooth surface. Thus, the appropriate application time and formulation characteristics of AP-MFP will also need to be determined in future studies. 

## 5. Conclusions

The application of AP-MFP was shown to significantly reduce substantial defects, ΔZ, and Ld and maintain Vickers hardness compared with the conventional APF application method, as well as to improve acid resistance both qualitatively and quantitatively. Furthermore, AP-MFP promoted the formation of CaF_2_, leading to the homogenous aggregation of fine particles on the surface of the dentin, sealing dentin tubules, and enhancing acid resistance in their vicinity. These results indicate that the AP-MFP-based fluoride application method suppresses the demineralization caused by citric acid and could emerge as a new professional-care method for preventing acid-induced erosion that is more powerful and biologically safer than the conventional method.

## Figures and Tables

**Figure 1 materials-16-05230-f001:**
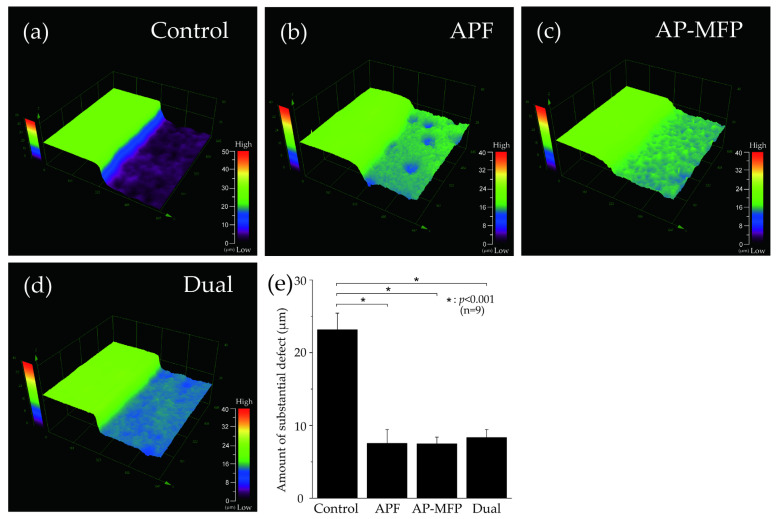
Step height profiles measured using 3D laser microscopy. Boundary images of the reference and experimental surfaces (RS and ES, respectively) after pH cycling in the (**a**) control (no fluoride), (**b**) APF, (**c**) AP-MFP, and (**d**) dual groups. In panels (**a**–**d**), the RS, which was protected using wax and was therefore not demineralized, is seen on the left, and the ES, which was not protected and was therefore demineralized, is seen on the right. (**e**) Graphical representation of the defect sizes after demineralization. *n* = 9 per group; * *p* < 0.001.

**Figure 2 materials-16-05230-f002:**
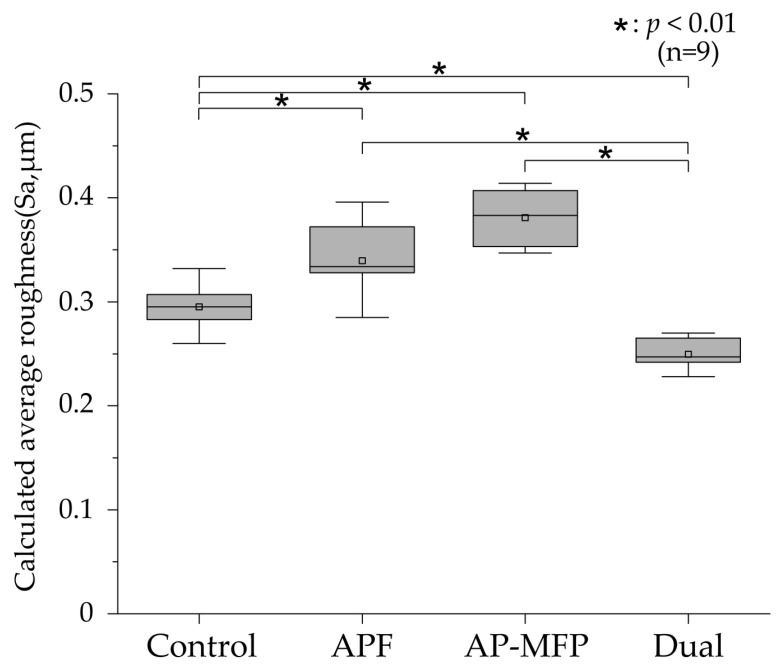
Calculated average roughness values after pH cycling. The median value is indicated by the horizontal line in the middle of each box, and the lower and upper boundaries indicate the 25th (Q1) and 75th (Q3) percentiles, respectively. Gray squares indicate mean values. *n* = 9 per group; * *p* < 0.01.

**Figure 3 materials-16-05230-f003:**
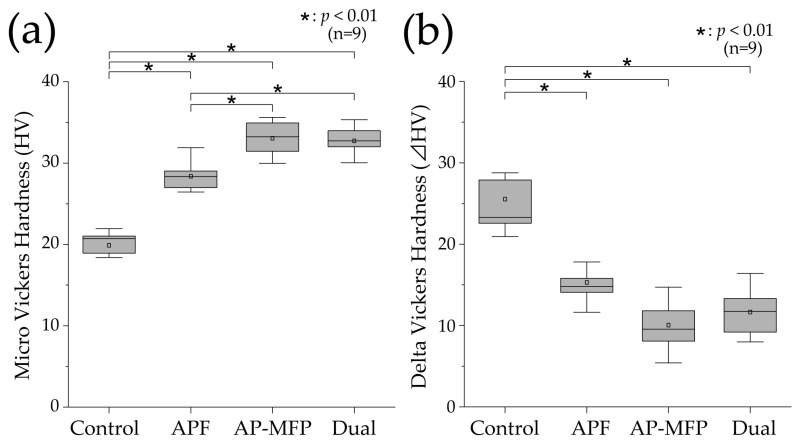
(**a**) Vickers hardness (HV) values after pH cycling. (**b**) ΔHV values (difference in the HV values between the RS and ES) after pH cycling. The median value is indicated by the horizontal line in the middle of each box, and the lower and upper boundaries indicate the 25th (Q1) and 75th (Q3) percentiles, respectively. The white squares indicate the mean value. *n* = 9 per group; * *p* < 0.01.

**Figure 4 materials-16-05230-f004:**
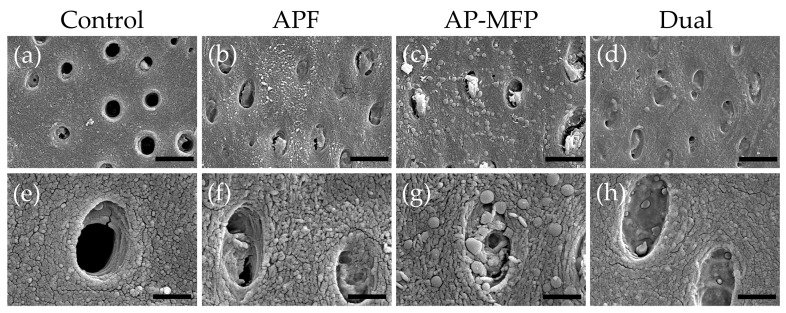
Scanning electron microscopy images of the dentin surface after pH cycling in the control (**a**,**e**), APF (**b**,**f**), AP-MFP (**c**,**g**), and dual (**d**,**h**) groups. (**a**–**d**) Scale bar: 5 μm. All images were recorded at 5000-fold magnification; carbon deposition sample. (**e**–**h**) Scale bar: 1.25 μm. All images were recorded at 15,000-fold magnification; carbon deposition sample.

**Figure 5 materials-16-05230-f005:**
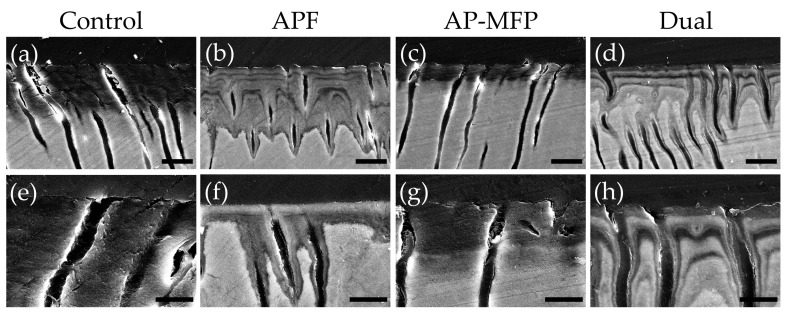
Scanning electron microscopy images of dentin cross-sections after pH cycling from the control (**a**,**e**), APF (**b**,**f**), AP-MFP (**c**,**g**), and dual (**d**,**h**) groups. (**a**–**d**) Scale bar: 10 μm. All images were acquired at 2500-fold magnification; carbon deposition sample. (**e**–**h**) Scale bar: 5 μm. All images were acquired at 5000-fold magnification; carbon deposition sample.

**Figure 6 materials-16-05230-f006:**
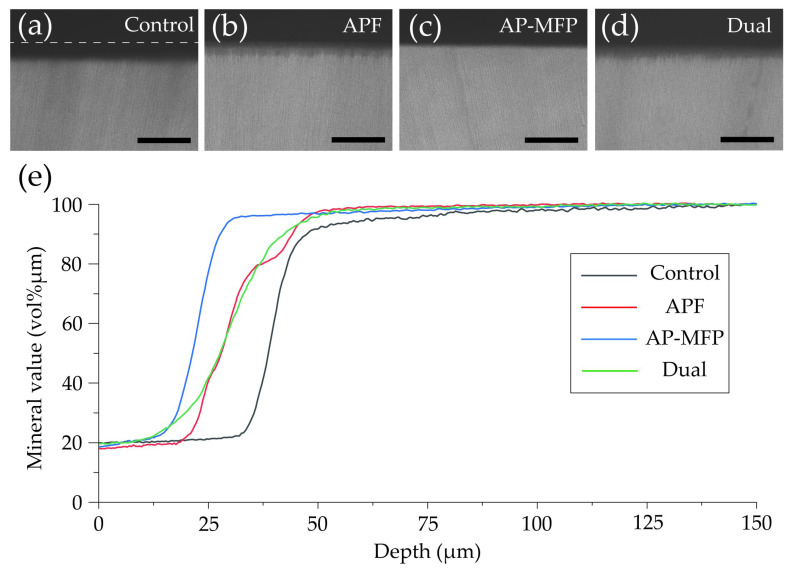
Contact microradiography images of dentin cross-sections after pH cycling from the (**a**) control, (**b**) APF, (**c**) AP-MFP, and (**d**) dual groups. Scale bar: 100 μm. (**e**) Graphical representation of the mineral values by depth. The black, red, blue, and green lines represent the control, APF, AP-MFP, and dual groups, respectively.

**Figure 7 materials-16-05230-f007:**
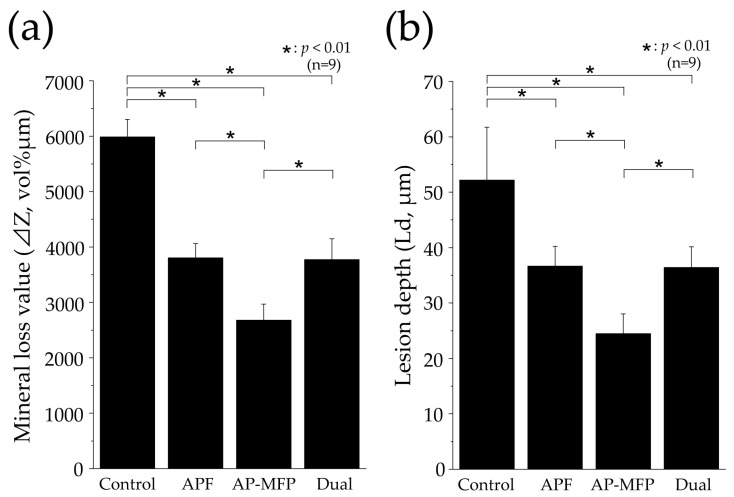
Graphical representation of (**a**) mineral loss (ΔZ) and (**b**) lesion depth (Ld) values after pH cycling. *n* = 9 per group, * *p* < 0.01. The depth of demineralization was measured from the surface prior to the demineralization experiment up to a site with 95% healthy dentin.

## Data Availability

All data are included in the manuscript.

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
