# Peer review of "Inhibition of Citric Acid-Induced Dentin Erosion by an Acidulated Phosphate Sodium Monofluorophosphate Solution"

_materials, 2023, doi:10.3390/ma16155230_

Round 1
Reviewer 1 Report
Dear Authors,
greetings!
The manuscript “Inhibition of citric acid-induced dentin erosion by an acidulated phosphate sodium monofluorophosphate solution” is dedicated to compare the capacity of MFP and APF in preventing dentin erosion provoked by acid challenge.
The study was well performed and the manuscript well written. However, there are some aspects that still need attention:
1) In the “Abstract” it is necessary to present the meaning of the abbreviation “AP”.
2) In “Materials and Methods” section, it is necessary to provide information regarding the supplier of chemicals used, and protocol applied to prepare solutions, as same as final molarities of phosphate fluoride and sodium monofluorophosphate. The study also lacks experiments of surface analysis X-ray Photoelectron Spectroscopy (XPS), presenting information on fluoride ion 1s orbit (intensity (cps) versus binding energy(eV)). It is an important experiment especially to the discussion on mechanism.
3) After performing XPS analysis it is interesting to review “Discussion” and “Conclusion” sections considering the results obtained.
Author Response
The manuscript “Inhibition of citric acid-induced dentin erosion by an acidulated phosphate sodium monofluorophosphate solution” is dedicated to compare the capacity of MFP and APF in preventing dentin erosion provoked by acid challenge.
The study was well performed and the manuscript well written. However, there are some aspects that still need attention:
> We strongly appreciate the reviewer's comment. We are thankful for the time and energy you expended.
1) In the “Abstract” it is necessary to present the meaning of the abbreviation “AP”.
> The reviewer's comment is correct. In accordance with the reviewer's comment, we have added abstract as described below.
Page 1, Line 17-19
Bovine dentin was used, and four groups were created: i) APF (9000 ppmF, pH 3.6) 4 min group, ii) acidulated phosphate MFP (AP-MFP,9000 ppmF, pH 3.6) 4 min group, iii) AP-MFP 2 min + APF 2 min (dual) group, and iv) no fluoride application (control) group.
2) In “Materials and Methods” section, it is necessary to provide information regarding the supplier of chemicals used, and protocol applied to prepare solutions, as same as final molarities of phosphate fluoride and sodium monofluorophosphate.
> We appreciate the reviewer's comment on this point. In accordance with the reviewer's comment, we have added materials and method as described below.
Page 3, Line 118-123
Citrate buffer solution was prepared by mixing 6.72 g of citric acid (7447-40-7, FUJIFILM Wako, Osaka, Japan) and 4.41 g of sodium citrate (68-04-2, FUJIFILM Wako, Osaka, Japan) in 1000 mL. HEPES buffer is composed of 0.1M CaCl2 (10043-52-4, FU-JIFILM Wako, Osaka, Japan) 30 mL, 0.1M KH2PO4 (7778-77-0, FUJIFILM Wako, Osaka, Japan) 18 mL, KCl (7447-40-7, FUJIFILM Wako, Osaka, Japan) 9.69 g, HEPES (7365-45-9, FUJIFILM Wako, Osaka, Japan) 4.77 g was mixed up to 1000 mL and adjusted to pH 7.3.
The study also lacks experiments of surface analysis X-ray Photoelectron Spectroscopy (XPS), presenting information on fluoride ion 1s orbit (intensity (cps) versus binding energy(eV)). It is an important experiment especially to the discussion on mechanism.
> We appreciate for suggesting the X-ray Photoelectron Spectroscopy (XPS) experiment, The reviewer's comment is correct. AP-MFP was developed by us, and this is the first paper with data on its action on dentin. Therefore, there are no previous studies on the effect and mechanism of AP-MFP and XPS analysis. We intend to summarize the results of XPS, atomic mapping (EPMA), and XRD analysis for the reaction and mechanism between AP-MFP and dentin for the content of the next paper. We have already done atomic mapping (EPMA), however it is difficult to include all of them in this paper because we have not been able to secure the research budget to do XPS analysis.
3) After performing XPS analysis it is interesting to review “Discussion” and “Conclusion” sections considering the results obtained.
> Thanks for your interest in the results. Although it will not be ready in time for this paper, we hope to compile atomic mapping images (Ca, P, F) after AP-MFP on dentin, XRD analysis after AP-MFP on HAP powder and XPS analysis of dentin for publication in the next paper.
Again, thank you for giving us the opportunity to strengthen our manuscript with your valuable comments and queries. We have worked hard to incorporate your feedback and hope that these revisions persuade you to accept our submission.
Reviewer 2 Report
Dear Authors, this paper about the inhibition of citric acid induced dentin erosion by an acidulated phosphate sodium monofluorophosphate solution is really interesting and well performed. I am pretty sure that it will help both scientists and dental professionals to improve their work.
Some small issues need to be solved before its final publication in the Journal.
Abstract: please, divide it into "introduction", "material and methods", "results", "conclusions".
Introdcution: this is a really important part od an article and it helps readers to deep into the subject of your study. You need to improve this part adding a small chapter about methods of dental protection against demineralization. This paper could be helpful: Ludovichetti FS, Signoriello AG, Colussi N, Zuccon A, Stellini E, Mazzoleni S. Soft drinks and dental erosion during pediatric age: a clinical investigation. Minerva Dent Oral Sci. 2022 Oct;71(5):262-269.
Materials and methods and results part are well performed and easy to understand.
Discussion:
The discussion should be more specific. The current discussion is somewhat general, and it does not provide enough detail about the mechanisms by which AP-MFP improves the acid resistance of dentin. The author could improve the specificity of the discussion by providing more information about the chemical reactions that occur when AP-MFP is applied to dentin.
The current discussion does not provide any critical evaluation of the study's findings. The author could improve the criticality of the discussion by discussing the limitations of the study and by suggesting areas for future research.
Author Response
Dear Authors, this paper about the inhibition of citric acid induced dentin erosion by an acidulated phosphate sodium monofluorophosphate solution is really interesting and well performed. I am pretty sure that it will help both scientists and dental professionals to improve their work.
> We strongly appreciate the reviewer's comment. We are thankful for the time and energy you expended.
Some small issues need to be solved before its final publication in the Journal.
> In accordance with the reviewer's comment, we have added abstract as described below.
Page 1, Line 23-24
The ΔZ value of the AP-MFP group was 2679 ± 290.2 vol% μm, significantly smaller than the APF group's 3806 ± 257.5 vol% μm (p<0.01).
> We checked again the author guideline of the materials and the most recent paper that came online, and no subsections were provided. Due to the abstract number of words, we did not fill in the measured values of all results, but a short summary of the results is written before the conclusion.
Page 1, Line 19-23
Compared to the conventional APF application method, the application of AP-MFP was shown to significantly reduce substantial defects, mineral loss, and lesion depth; better maintain Vickers hardness; and promote homogenous aggregation of fine CaF2 particles to seal dentin tubules, enhancing acid resistance in their vicinity.
Introdcution: this is a really important part od an article and it helps readers to deep into the subject of your study. You need to improve this part adding a small chapter about methods of dental protection against demineralization. This paper could be helpful: Ludovichetti FS, Signoriello AG, Colussi N, Zuccon A, Stellini E, Mazzoleni S. Soft drinks and dental erosion during pediatric age: a clinical investigation. Minerva Dent Oral Sci. 2022 Oct;71(5):262-269.
> The reviewer's comment is correct. In accordance with the reviewer's comment, we have added introduction part and reference as described below.
Page 2, Line 57-60
Toothpastes containing fluoride have not proved effective in preventing erosion and no significant correlations were found between the type of toothbrush used and its frequency of use in 231 children [6].
Reference No.6
Ludovichetti FS, Signoriello AG, Colussi N, Zuccon A, Stellini E, Mazzoleni S. Soft drinks and dental erosion during pediatric age: a clinical investigation. Minerva Dent Oral Sci. 2022 Oct;71(5):262-269.
Materials and methods and results part are well performed and easy to understand.
> We appreciate the reviewer's comment on this point.
Discussion:
The discussion should be more specific. The current discussion is somewhat general, and it does not provide enough detail about the mechanisms by which AP-MFP improves the acid resistance of dentin. The author could improve the specificity of the discussion by providing more information about the chemical reactions that occur when AP-MFP is applied to dentin.
> We appreciate the reviewer's comment on this point. There is a reason why the discussion is only general: AP-MFP is new agent we have developed, and this is the first time it has been applied to dentin. Therefore, prior studies have only provided enamel data, and the detailed mechanism of the reaction of AP-MFP with dentin is unknown; the chemical reaction between MFP and dentin is represented by the references cited, and further detailed mechanisms will have to await future studies. Following the reviewer's comments, we have added a discussion of what we consider dentin to discussion.
Page 11, Line 340-343
Compared to enamel, dentin has a porous structure characterized by the presence of tubules and a higher proportion of collagen, and this composition offers several advantages in terms of ion permeability and reaction kinetics [4,24].
The current discussion does not provide any critical evaluation of the study's findings. The author could improve the criticality of the discussion by discussing the limitations of the study and by suggesting areas for future research.
> The reviewer's comment is correct. In accordance with the reviewer's comment, we have added discussion section as described below.
Page 14, Line 483-488
Furthermore, it should be noted that this study was conducted in an in vitro setting in the absence of saliva and plaque, which may lead to different behaviors in the actual oral environment where saliva proteins and enzymes are abundant. Therefore, further investigations are also warranted to assess the oral stability of AP-MFP in the presence of MFP-degrading enzymes. Future experiments should be conducted with human teeth.
Again, thank you for giving us the opportunity to strengthen our manuscript with your valuable comments and queries. We have worked hard to incorporate your feedback and hope that these revisions persuade you to accept our submission.
Round 2
Reviewer 1 Report
The manuscript was improved as requested.